# Barking up the right tree: an approach to search over molecule synthesis DAGs

**John Bradshaw**
University of Cambridge
MPI for Intelligent Systems
jab255@cam.ac.uk

**Brooks Paige**
University College London
The Alan Turing Institute
b.paige@ucl.ac.uk

**Matt J. Kusner**
University College London
The Alan Turing Institute
m.kusner@ucl.ac.uk

**Marwin H. S. Segler**
WWU Münster
Microsoft Research Cambridge, UK
marwin.segler@wwu.de

**José Miguel Hernández-Lobato**
University of Cambridge
The Alan Turing Institute
Microsoft Research Cambridge, UK
jmh233@cam.ac.uk

## Abstract

When designing new molecules with particular properties, it is not only important what to make but crucially *how to make it*. These instructions form a synthesis directed acyclic graph (DAG), describing how a large vocabulary of simple building blocks can be recursively combined through chemical reactions to create more complicated molecules of interest. In contrast, many current deep generative models for molecules ignore synthesizability. We therefore propose a deep generative model that better represents the real world process, by directly outputting molecule synthesis DAGs. We argue that this provides sensible inductive biases, ensuring that our model searches over the same chemical space that chemists would also have access to, as well as interpretability. We show that our approach is able to model chemical space well, producing a wide range of diverse molecules, and allows for unconstrained optimization of an inherently constrained problem: maximize certain chemical properties such that discovered molecules are synthesizable.

## 1 Introduction

Designing and discovering new molecules is key for addressing some of world's most pressing problems: creating new materials to tackle climate change, developing new medicines, and providing agrochemicals to the world's growing population. The molecular discovery process usually proceeds in design-make-test-analyze cycles, where new molecules are designed, made in the lab, tested in lab-based experiments to gather data, which is then analyzed to inform the next design step [29].

There has been a lot of recent interest in accelerating this feedback loop, by using machine learning (ML) to design new molecules [3, 14, 23, 35, 38, 39, 52, 64, 67, 70, 71, 82]. We can break up these developments into two connected goals: G1. **Learning strong generative models of molecules** that can be used to sample novel molecules, for downstream screening and scoring tasks; and G2. **Molecular optimization**: Finding molecules that optimize properties of interest (e.g. binding affinity, solubility, non-toxicity, etc.). Both these goals try to reduce the total number of design-make-test-analyze cycles needed in a complementary manner, G1 by broadening the number of molecules that can be considered in each step, and G2 by being smarter about which molecule to pick next.

Significant progress towards these goals has been made using: (a) representations ranging from strings and grammars [23, 44, 67], to fragments [35, 53] and molecular graphs [46, 47, 72]; and (b)

model classes including latent generative models [16, 23, 44, 49] and autoregressive models [46, 67]. However, in these approaches to molecular design, there is no explicit indication that the designed molecules can actually be synthesized; the next step in the design-make-test-analyze cycle and the prerequisite for experimental testing. This hampers the application of computer-designed compounds in practice, where rapid experimental feedback is essential. While it is possible to address this with post-hoc synthesis planning [21, 32, 65, 68, 73], this is unfortunately very slow.

One possibility to better link up the design and make steps, in ML driven molecular generation, is to explicitly include synthesis instructions in the design step. Based on similar ideas to some of the discrete enumeration algorithms established in chemoinformatics, which construct molecules from building blocks via virtual chemical reactions [11, 27, 31, 75, 76], such models were recently introduced [7, 43]. Bradshaw et al. [7] proposed a model to choose which reactants to combine from an initial pool, and employed an autoencoder for optimization. However, their model was limited to single-step reactions, whereas most molecules require multi-step syntheses (e.g. see Figure 1). Korovina et al. [43] performed a random walk on a reaction network, deciding which molecules to assess the properties of using Bayesian optimization. However, this random walk significantly limits the ability of that model to optimize molecules for certain properties.[1]

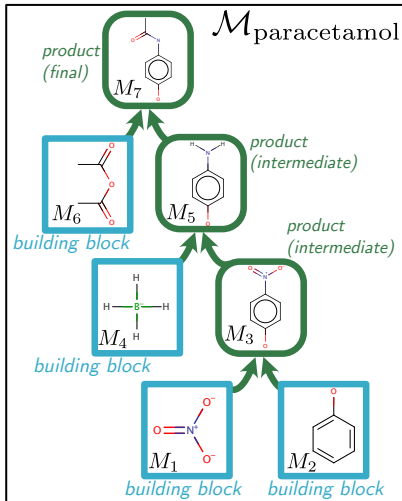

Figure 1: An example synthesis DAG for paracetamol [19] Note that we are ignoring some reagents, conditions and details of chirality for simplicity.

In this work to address this gap, we present a new architecture to generate *multi-step molecular synthesis routes* and show how this model can be used for targeted optimization. We (i) explain how one can represent synthetic routes as directed acyclic graphs (DAGs), (ii) propose a novel hierarchical neural message passing procedure that exchanges information among multiple levels in such a synthesis DAG, and (iii) develop an efficient serialization procedure for synthesis DAGs. Finally, we show how our approach leads to encoder and decoder networks which can both be integrated into widely used architectures and frameworks such as latent generative models (G1) [42, 59, 74], or reinforcement learning-based optimization procedures to sample and optimize (G2) novel molecules along with their synthesis pathways. Compared with models not constrained to also generate synthetically tractable molecules, competitive results are obtained.

## 2 Formalizing Molecular Synthesis

We begin by formalizing a multi-step reaction synthesis as a DAG (see Figure 1).[2]

**Synthesis pathways as DAGs**  At a high level, to synthesize a new molecule $M_T$, one needs to perform a series of reaction steps. Each reaction step takes a set of molecules and physically combines them under suitable conditions to produce a new molecule. The set of molecules that react together are selected from a pool of molecules that are available at that point. This pool consists of a large set of initial, easy-to-obtain starting molecules (building blocks) and the intermediate products already created. The reaction steps continue in this manner until the final molecule $M_T$ is produced, where we make the assumption that all reactions are deterministic and produce a single primary product.

For example, consider the synthesis pathway for paracetamol, in Figure 1, which we will use as a running example to illustrate our approach. Here, the initial, easy-to-obtain building block molecules

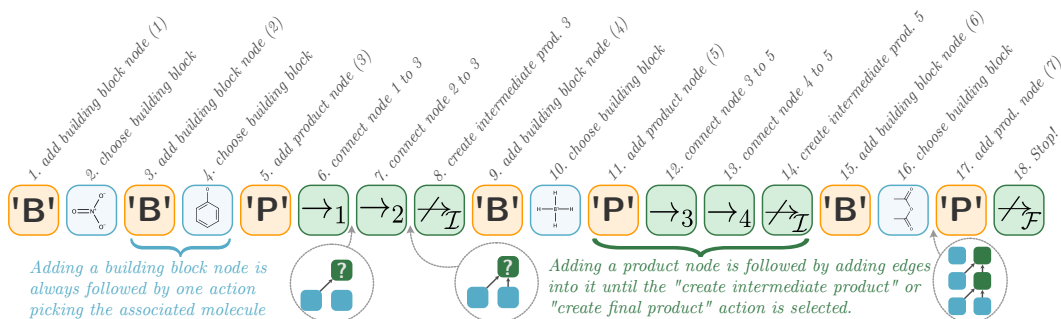

Figure 2: An example of how we can *serialize* the construction of the DAG shown in Figure 1, with the corresponding DAG at that point in the sequence shown for three different time-points in the grey circles. The serialized construction sequence consists of a sequence of actions. These actions can be classified into belonging to three different types: (A1) node addition, (A2) building block molecular identity, and (A3) connectivity choice. By convention we start at the building block node that is furthest from the final product node, sampling randomly when two nodes are at equivalent distances.

are shown in blue boxes. Reactions are signified with arrows, which produce a product molecule (outlined in green) from reactants. These product molecules can then become reactants themselves.

In general, this multi-step reaction pathway forms a synthesis DAG, which we shall denote $\mathcal{M}$. Specifically, note that it is directed from reactants to products, and it is not cyclic, as we do not need to consider reactions that produce already obtained reactants.

## 3 Our Models

Here we describe a generative model of synthesis DAGs. This model can be used flexibly as part of a larger architecture or model, as shown in the second half of this section.

### 3.1 A probabilistic generative model of synthesis DAGs

We begin by first describing a way to serialize the construction of synthesis DAGs, such that a DAG can be iteratively built up by our model predicting a series of actions. We then describe our model, which parameterizes probability distributions over each of these actions. Further details including specific hyperparameters along with a full algorithm are deferred to the Appendix.[3]

**Serializing the construction of DAGs** Consider again the synthesis DAG for paracetamol shown in Figure 1; how can the actions necessary for constructing this DAG be efficiently ordered into a sequence to obtain the final molecule? Figure 2 shows such an approach. Specifically, we divide actions into three types: A1. *Node-addition (shown in yellow)*: What type of node (building block or product) should be added to the graph?; A2. *Building block molecular identity (in blue)*: Once a building block node is added, what molecule should this node represent?; A3. *Connectivity choice (in green)*: What reactant nodes should be connected to a product node (i.e., what molecules should be reacted together)?

As shown in Figure 2 the construction of the DAG then happens through a sequence of these actions. Building block ('B') or product nodes ('P') are selected through action type A1, before the identity of the molecule they contain is selected. For building blocks this consists of choosing the relevant molecule through an action of type A2. Product nodes' molecular identity is instead defined by the reactants that produce them, therefore action type A3 is used repeatedly to either select these incoming reactant edges, or to decide to form an intermediate ($\not\rightarrow_\mathcal{I}$) or final product ($\not\rightarrow_\mathcal{F}$). In forming a final product all the previous nodes without successors are connected up to the final product node, and the sequence is complete.

In creating a DAG, $\mathcal{M}$, we will formally denote this sequence of actions, which fully defines its structure, as $\mathcal{M} = [V^1, V^2, V^3, \ldots, V^L]$. We shall denote the associated action types as $\boldsymbol{A} = [A^1, A^2, A^3, \ldots, A^L]$ with $A^l \in \{A1, A2, A3\}$, and note that these are also fully defined by the

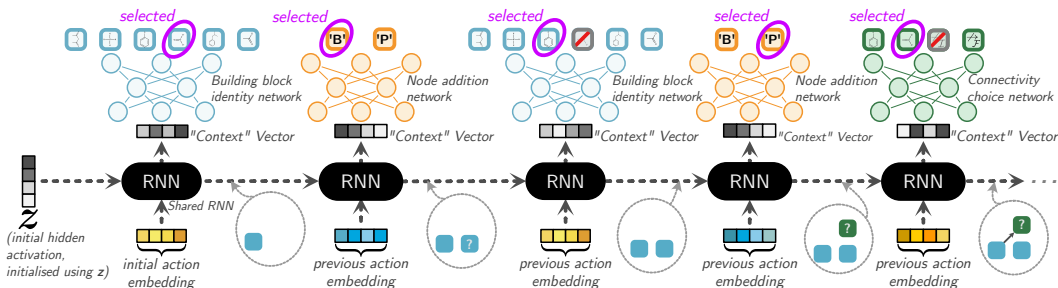

Figure 3: A depiction of how we use neural networks to parameterize the probability of picking actions at stages 1-6 of Figure 2 (note that as stage 1 always suggests a building block node it is automatically completed). A shared RNN for the different action networks receives an embedding of the previous action chosen and creates a context vector for the action network. When using our model as part of an autoencoder network then the initial hidden layer is parameterized by the latent space sample, $z$. Each type of action network chooses a subsequent action to take (with actions that are impossible being masked out, such as selecting an already existing building block or creating an intermediate product before connecting up any reactants). The process continues until the 'create final product' node is selected (see Figure 3 in the Appendix). Graph neural networks are used for computing embeddings of molecules where required.

previous actions chosen (e.g. after choosing a building block identity you always go back to adding a new node). The molecules corresponding to the nodes produced so far we will denote as $M_i$, with $i$ referencing the order in which they are created. We will also abuse this notation and use $M_{<l}$ to describe the set of molecule nodes existing at the time of predicting action $l$. Finally, we shall denote the set of initial, easy-to-obtain building blocks as $\mathcal{R}$.

**Defining a probabilistic distribution over construction actions**    We are now ready to define a probabilistic distribution over $\mathcal{M}$. We allow our distribution to depend on a latent variable $z \in \mathbb{R}^d$, the setting of which we shall ignore for now and discuss in the sections that follow. We propose an auto-regressive factorization over the actions:

$$p_\theta(\mathcal{M}|z) = \prod_{l=1}^{L} p_\theta(V_l|V_{<l}, z) \tag{1}$$

Each $p_\theta(V_l|V_{<l}, z)$ is parameterized by a neural network, with weights $\theta$. The structure of this network is shown in Figure 3. It consists of a shared RNN module that computes a 'context' vector. This 'context' vector then gets fed into a feed forward action-network for predicting each action. A specific action-network is used for each action type, and the action type also constrains the actions that can be chosen from this network: $V_{|A^l=\text{A1}}^l \in \{\text{'B', 'P'}\}$, $V_{|A^l=\text{A2}}^l \in \mathcal{R}$, and $V_{|A^l=\text{A3}}^l \in M_{<l} \cup \{\not\rightarrow_\mathcal{I}, \not\rightarrow_\mathcal{F}\}$. The initial hidden layer of the RNN is initialized by $z$ and we feed in the previous chosen action embedding as input to the RNN at each step. Please see the Appendix for further details including pseudocode for the entire generative procedure in detail as a probabilistic program (Alg. 1).

**Action embeddings**    For representing actions to our neural network we need continuous embeddings, $h \in \mathbb{R}^d$. Actions can be grouped into either (i) those that *select a molecular graph* (be that a building block, $g \in \mathcal{R}$, or a reactant already created, $g' \in M_{<l}$); or (ii) those that perform a more *abstract action* on the DAG, such as creating a new node ('B', 'P'), producing an intermediate product ($\not\rightarrow_\mathcal{I}$), or lastly producing a final product ($\not\rightarrow_\mathcal{F}$). For the abstract actions the embeddings we use are each distinct learned vectors that are parameters of our model.

With actions involving a molecular graph, instead of learning embeddings for each molecule we compute embeddings that take the structure of the molecular graph into account. Perhaps the simplest way to do this would be using fixed molecule representations such as Morgan fingerprints [50, 60]. However, Morgan fingerprints, being fixed, cannot learn which characteristics of a molecule are important for our task. Therefore, for computing molecular embeddings we instead choose to use deep graph neural networks (GNN), specifically Gated Graph Neural Networks [45] (GGNNs). GNNs can learn which characteristics of a graph are important and have been shown to perform well on a variety of tasks involving small organic molecules [18, 22].

**Reaction prediction** When forming an intermediate or final product we need to obtain the molecule that is formed from the reactants chosen. At training time we can simply fill in the correct molecule using our training data. However at test time this information is not available. We assume however we have access to an oracle, Product($\cdot$), that given a set of reactants produces the major product formed (or randomly selects a reactant if the reaction does not work). For this oracle we could use any reaction predictor method [6, 12, 13, 17, 20, 34, 40, 41, 63, 66, 79]. In our experiments we use the Molecular Transformer [63], as it currently obtains state-of-the-art performance on reaction prediction [63, Table 4]. Further details of how we use this model are in the Appendix.

It is worth pointing out that current reaction predictor models are not perfect, sometimes predicting incorrect reactions. By treating this oracle as a black-box however, our model can take advantage of any future developments of these methods or use alternative predictors for other reasons, such as to control the reaction classes used.

## 3.2 Variants of our model

Having introduced our general model for generating synthesis DAGs of (molecular) graphs (DoGs), we detail two variants: an autoencoder (DoG-AE) for learning continuous embeddings (G1), and a more basic generator (DoG-Gen) for performing molecular optimization via fine-tuning or reinforcement learning (G2).

### 3.2.1 DoG-AE: Learning a latent space over synthesis DAGs

For G1, we are interested in learning mappings between molecular space and latent continuous embeddings. This latent space can then be used for exploring molecular space via interpolating between two points, as well as sampling novel molecules for downstream screening tasks. To do this we will use our generative model as the *decoder* in an autoencoder structure, which we shall call DoG-AE. We specify a Gaussian prior over our latent variable $z$, where each $z \sim p(z)$ can be thought of as describing different types of synthesis pathways.

The autoencoder consists of a stochastic mapping from $\mathbf{z}$ to synthesis DAGs and a separate *encoder*, $q_\phi(\mathbf{z} \mid \mathcal{M})$ a stochastic mapping from synthesis DAGs, $\mathcal{M}$, to latent space. Using our generative model of synthesis DAGs, described in the previous subsection, as our decoder (with the latent variable initializing the hidden state of the RNN as shown in Figure 3), we are left to define both our autoencoder loss and encoder.

**Autoencoder loss** We choose to optimize our autoencoder model using a Wasserstein autoencoder (WAE) objective with a negative log likelihood cost function [74],

$$\min_{\phi,\theta} \mathbb{E}_{\mathcal{M}\sim p(\mathcal{M})}\mathbb{E}_{q_\phi(\mathbf{z}|\mathcal{M})}\Big[ -\log p_\theta(\mathcal{M} \mid \mathbf{z})\Big] + \lambda\mathcal{D}(q_\phi(\mathbf{z}), p(\mathbf{z})), \qquad (2)$$

where following Tolstikhin et al. [74] $\mathcal{D}(\cdot, \cdot)$ is a maximum mean discrepancy (MMD) divergence measure [25] and $\lambda = 10$; alternative autoencoder objectives could easily be used instead.

**Encoder** At a high level, our encoder consists of a two-step hierarchical message passing procedure described in Figure 4: (1) Molecular graph message passing; (2) Synthesis graph message passing. Given a synthesis DAG $\mathcal{M}$, each molecule $M$ within that pathway can itself be represented as a graph: each atom $a$ is a node and each bond $b_{a,a'}$ between atoms $a$ and $a'$ is an edge. As we did for the decoder, we can embed these molecular graphs into continuous space using any graph neural network, in practice we choose to use the same network as the decoder (also sharing the same weights).

Given these initial molecular graph embeddings $\mathbf{e}_M^0$ (now node embeddings for the DAG), we would like to use them to embed the entire synthesis DAG $\mathcal{M}$. We do so by passing the molecular embeddings through another second step of message passing (the synthesis graph message passing), this time across the synthesis DAG. Specifically, we use the message function $g_m$ to compute messages as $\mathbf{m}_M^t = g_m(\mathbf{e}_M^{t-1}, \mathbf{e}_{N(M)}^{t-1})$, where $N(M)$ are the predecessor molecules in $\mathcal{M}$ (i.e., the reactants used to create $M$), before updating the DAG node embeddings with this message using the function $\mathbf{e}_M^t = g_e(\mathbf{m}_M^t, \mathbf{e}_M^{t-1})$. After $T'$ iterations we have the final DAG node embeddings $\mathbf{e}_M^{T'}$, which we aggregate as $\boldsymbol{\mu} = g_\mu(\mathbf{E}_\mathcal{M}^{T'})$ and $\boldsymbol{\sigma}^2 = \exp(g_{\log\sigma^2}(\mathbf{E}_\mathcal{M}^{T'}))$. These parameterize the mean and variance of the encoder as: $q_\phi(\mathbf{z} \mid \mathcal{M}) := \mathcal{N}(\boldsymbol{\mu}, \boldsymbol{\sigma}^2)$.

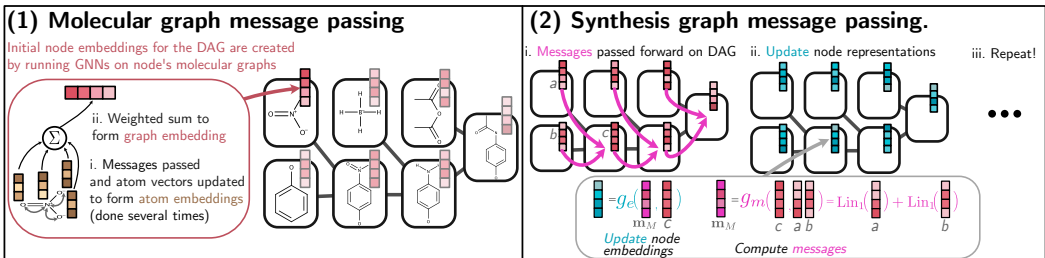

Figure 4: The encoder embeds the DAG of Graphs (DoG) into a continuous latent space. It does this using a two-step hierarchical message passing procedure. In step 1 (Molecular graph message passing) it computes initial embeddings for the DAG nodes by forming graph-level embeddings using a GNN on the molecular graph associated with each node. In step 2 (Synthesis graph message passing) a message-passing algorithm is again used, however, this time on the synthesis DAG itself, passing messages forward. In our experiments we use GGNNs [45] for both message passing steps (see the Appendix for further details). The final representation of the DAG is taken from the node embedding of the final product node.

We are again flexible about the architecture details of the message passing procedure (i.e., the architecture of the GNN functions $g_m, g_e, g_\mu, g_{\log \sigma^2}$). We leave these specifics up to the practitioner and describe in the Appendix details about the exact model we use in our experiments, where similar to the GNN used for the molecular embeddings, we use a GGNN [45] like architecture.

### 3.2.2 DoG-Gen: Molecular optimization via fine-tuning

For molecular optimization, we consider a model trained without a latent space; we use our probabilistic generator of synthesis DAGs and fix $\mathbf{z} = \mathbf{0}$, we call this model DoG-Gen. We then adopt the hill-climbing algorithm from Brown et al. [8, §7.4.6] [51, 67], an example of the cross-entropy method [15], or also able to be viewed as a variant of REINFORCE [80] with a particular reward shaping (other RL algorithms could also be considered here). For this, our model is first pre-trained via maximum likelihood to match the training dataset distribution $p(\mathcal{M})$. For optimization, we can then fine-tune the weights $\theta$ of the decoder: this is done by sampling a large number of candidate DAGs from the model, ranking them according to a target, and then fine-tuning our model's weights on the top K samples (see Alg. 2 in the Appendix for full pseudocode of this procedure).

## 4 Experiments

We now evaluate our approach to generating synthesis DAGs on two connected goals set out in the introduction: (G1) can we model the space of synthesis DAGs well and using DoG-AE interpolate within that space, and (G2) can we find optimized molecules for particular properties using DoG-Gen and the fine-tuning technique set out in the previous section. To train our models, we create a dataset of synthesis DAGs based on the USPTO reaction dataset [48]. We detail the construction of this dataset in the Appendix (§ C.1). We train both our models on the same dataset and find that DoG-AE obtains a reconstruction accuracy (on our held out test set) of 65% when greedily decoding (i.e. picking the most probable action at each stage of decoding).

### 4.1 Generative modeling of synthesis DAGs

We begin by assessing properties of the *final* molecules produced by our generative model of synthesis DAGs. Ignoring the synthesis allows us to compare against previous generative models for molecules including SMILES-LSTM (a character-based autoregressive language model for SMILES strings)[67], the Character VAE (CVAE) [23], the Grammar VAE (GVAE) [44], the GraphVAE [72], the Junction Tree Autoencoder (JT-VAE) [35], the Constrained Graph Autoencoder (CGVAE) [47], and Molecule Chef [7].[4] Further details about the baselines are in the Appendix. These models cover a wide range of approaches for modeling structured molecular graphs. Aside from Molecule Chef, which is limited

Table 1: Table showing the percentage of valid molecules generated and then conditioned on this the uniqueness, novelty and normalized quality [8, §3.3] (all as %, higher better) as well as FCD score (Fréchet ChemNet Distance, lower better) [54]. For each model we generate the molecules by decoding from 20k prior samples from the latent space.

| Model Name | Validity (↑) | Uniqueness (↑) | Novelty (↑) | Quality (↑) | FCD (↓) |
|---|---|---|---|---|---|
| DoG-AE | 100.0 | 98.3 | 92.9 | 95.5 | 0.83 |
| DoG-Gen | 100.0 | 97.7 | 88.4 | 101.6 | 0.45 |
| Training Data | 100.0 | 100.0 | 0.0 | 100.0 | 0.21 |
| SMILES LSTM [67] | 94.8 | 95.5 | 74.9 | 101.93 | 0.46 |
| CVAE [23] | 96.2 | 97.6 | 76.9 | 103.82 | 0.43 |
| GVAE [44] | 74.4 | 97.8 | 82.7 | 98.98 | 0.89 |
| GraphVAE [72] | 42.2 | 57.7 | 96.1 | 94.64 | 13.92 |
| JT-VAE [35] | 100.0 | 99.2 | 94.9 | 102.34 | 0.93 |
| CGVAE [47] | 100.0 | 97.8 | 97.9 | 45.64 | 14.26 |
| Molecule Chef [7] | 98.9 | 96.7 | 90.0 | 99.0 | 0.79 |

to one step reactions and so is unable to cover as much of molecular space as our approach, these other baselines do not provide synthetic routes with the output molecule.

As metrics we report those proposed in previous works [7, 35, 44, 47, 67]. Specifically, validity measures how many of the generated molecules can be parsed by the chemoinformatics software RDKit [56]. Conditioned on validity, we consider the proportions of generated molecules that are unique (within the sample), novel (different to those in the training set), and pass the quality filters (normalized relative to the training set) proposed in Brown et al. [8, §3.3]. Finally we measure the Fréchet ChemNet Distance (FCD) [54] between generated and training set molecules. We view these metrics as useful sanity checks, showing that sensible molecules are produced, albeit with limitations [8], [57, §2]. We include additional checks in the Appendix.

Table 1 shows the results. Generally we see that many of these models perform comparably with no model performing better than all of the others on all of the tasks. The baselines JT-VAE and Molecule Chef have relatively high performance across all the tasks, although by looking at the FCD score it seems that the molecules that they produce are not as close to the original training set as those suggested by the simpler character based SMILES models, CVAE or SMILES LSTM. Encouragingly, we see that the models we propose provide comparable scores to many of the models that do not provide synthetic routes and achieve good performance on these sanity checks.

**The latent space of synthesis DAGs.** The advantage of our model over the others is that it directly generates synthesis DAGs, indicating how a generated molecule could be made. To visualize the latent space of DAGs we start from a training synthesis DAG and walk randomly in latent space until we have output five different synthesis DAGs. We plot the combination of these DAGs, which can be seen as a reaction network, in Figure 5. We see that as we move around the latent space many of the synthesis DAGs have subgraphs that are isomorphic, resulting in similar final molecules.

## 4.2 Optimizing synthesizable molecules

We next look at how our model can be used for the optimization of molecules with desirable properties. To evaluate our model, we compare its performance on a series of 10 optimization tasks from GuacaMol [8, §3.2] against the three best reported models Brown et al. [8, Table 2] found: (1) SMILES LSTM [67], which does optimization via fine-tuning; (2) GraphGA [33], a graph genetic algorithm (GA); and (3) SMILES GA [81], a SMILES based GA. We train all methods on the same data, which is derived from USPTO and, as such, should give a strong bias for synthesizability.

We note that we should not expect our model to find the best molecule if judged solely on the GuacaMol task score; our model has to build up molecules from set building blocks and pre-learned reactions, which although reflecting the real-life process of molecule design, means that it is operating in a more constrained regime. However, the final property score is not the sole factor that is important when considering a proposed molecule. Molecules also need to: (i) exist without degrading or

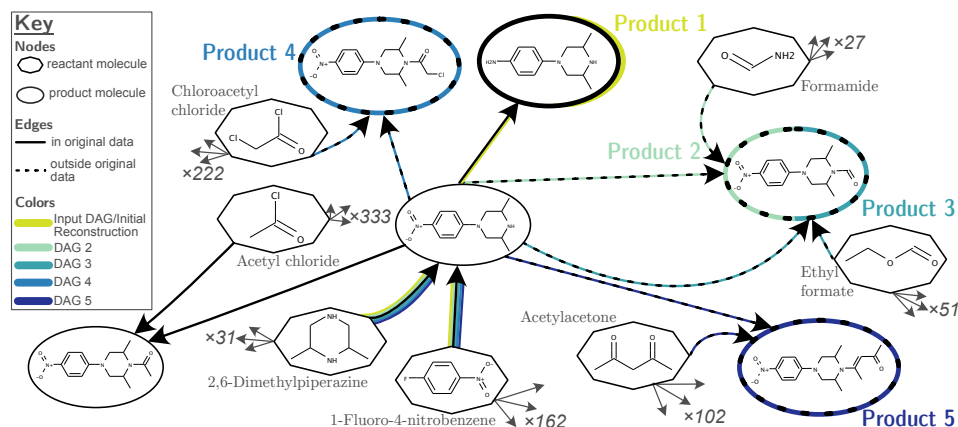

Figure 5: Using a variant of the DoG-AE model, as we randomly walk in the latent space we decode out to similar DAGs nearby, unseen in training. Reactions and nodes that exist in our original dataset are outlined in solid lines, whereas those that have been discovered by our model are shown with dashed lines.

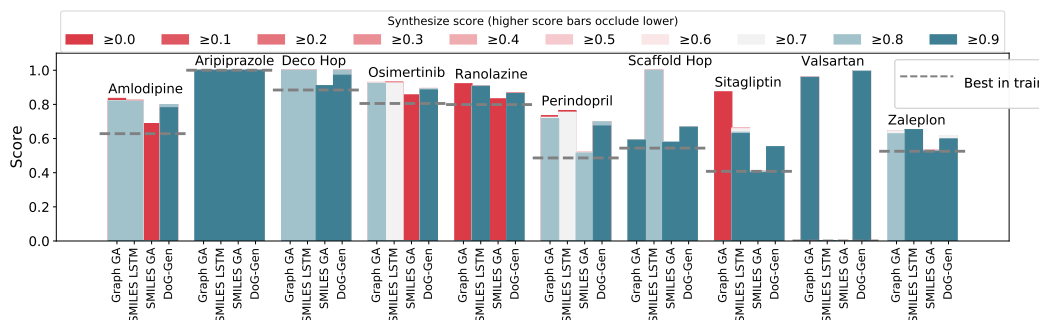

Figure 6: The score of the best molecule found by the different approaches over a series of ten GuacaMol benchmark tasks [8, §3.2], with the task name labeled above each set of bars. GuacaMol molecule scores (y-axis) range between 0 and 1, with 1 being the best. We also use colors to indicate the synthesizability score of the best molecule found. Note that bars representing a molecule within a higher synthesizability score bucket (e.g blue) will occlude lower synthesizability score bars (e.g. red). The dotted gray lines represent the scores of the best molecule in our training set.

reacting further (*i.e., be sufficiently stable*), and (ii) be able to actually be created in practice (*i.e., be synthesizable*). To quantify (i) we consider using the quality filters proposed in Brown et al. [8, §3.3]. To quantify (ii) we use Computer-Aided Synthesis Planning [4, 21, 68]. Specifically, we run a retrosynthesis tool on each molecule to see if a synthetic route can be found, and if so how many steps are involved [5]. We also measure an aggregated synthesizability score over each step (see § C.2 in the Appendix), with a higher synthesizability score indicating that the individual reactions are closer to actual reactions and so hopefully more likely to work. All results are calculated on the top 100 molecules found by each method for each GuacaMol task.

The results of the experiments are shown in Figures 6 and 7 and Table 2 (see the Appendix for further results). In Figure 6 we see that, disregarding synthesis, in general Graph GA and SMILES LSTM produce the best scoring molecules for the GuacaMol tasks. However, corroborating the findings of [21, FigureS6], we note that the GA methods regularly suggest molecules where no synthesis routes can be found. Our model, because it decodes synthesis DAGs, consistently finds high-scoring molecules while maintaining high synthesis scores. Furthermore, Figure 7 shows that a high fraction of molecules produced by our method pass the quality checks (consistent with the amount in the training data), whereas for some of the other methods the majority of molecules can fail.

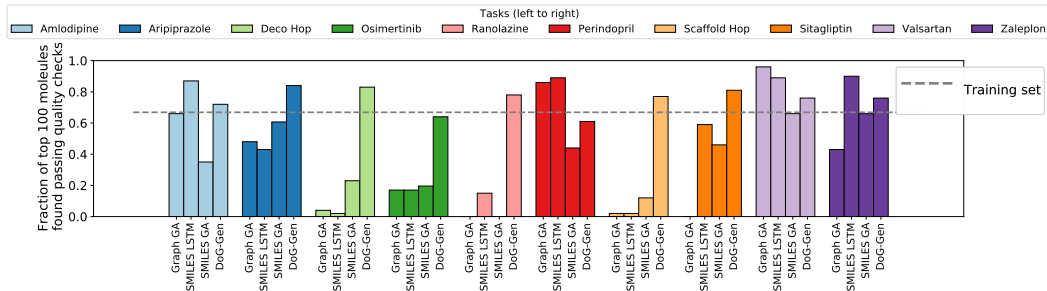

Figure 7: The fraction of the top 100 molecules proposed that pass the quality filters, over a series of ten GuacaMol tasks [8, §3.2]. The fraction of molecules in our initial training set (used for training the models) that pass the filters is shown by the dotted gray line.

Table 2: Table showing metrics quantifying the stability and synthesizability of the molecules suggested by each method for the GuacaMol optimization tasks. The metrics we use include: the fraction of molecules for which a synthetic route is found, the mean synthesizability score, the median number of synthesis steps (for synthesizable molecules), and the fraction of molecules that pass the quality filters from Brown et al. [8, §3.3]. The metrics are computed using the aggregation of the top 100 molecules for each GuacaMol task over all of the ten GuacaMol tasks we consider.

|  | Frac. Synthesizable (↑) | Synth. Score (↑) | Median # Steps (↓) | Quality (↑) |
|---|---|---|---|---|
| DoG-Gen | 0.9 | 0.76 | 4 | 0.75 |
| Graph GA | 0.42 | 0.33 | 6 | 0.36 |
| SMILES LSTM | 0.48 | 0.39 | 5 | 0.49 |
| SMILES GA | 0.29 | 0.25 | 3 | 0.39 |

## 5 Related Work

In computer-aided molecular discovery there are two main approaches for choosing the next molecules to evaluate, virtual screening [11, 55, 58, 69, 75, 77, 78] and de novo design [26, 62]. De novo design, whilst enabling the searching over large chemical spaces, can lead to unstable and unsynthesizable molecules [8, 21, 26]. Works to address this in the chemoinformatics literature have built algorithms around virtual reaction schemes [27, 76]. However, with these approaches there is still the problem of how best to search through this action space, a difficult discrete optimization problem, and so these algorithms often resorted to optimizing greedily one-step at a time or using GAs.

Neural generative approaches [23, 52, 67] have attempted to tackle this latter problem by designing models to be used with Bayesian optimization or RL based search techniques. However, these approaches have often lead to molecules containing unrealistic moieties [8, 21]. Extensions have investigated representing molecules with grammars [14, 44], or explicitly constrained graph representations [35–37, 47, 53, 61], which while fixing the realism somewhat, still ignores synthesizability.

ML models for generating or editing graphs (including DAGs) have also been developed for other application areas, such as citation networks, community graphs, network architectures or source code [2, 9, 10, 83, 84]. DAGs have been generated in both a bottom-up (e.g. [84]) and top-down (e.g. [10]) manner. Our approach is perhaps most related to Zhang et al. [84], which develops an autoencoder model for DAGs representing neural network architectures. However, synthesis DAGs have some key differences to those typically representing neural network architectures or source code, for instance nodes represent molecular graphs and should be unique within the DAG.

## 6 Conclusions

In this work, we introduced a novel neural architecture component for molecule design, which by directly generating synthesis DAGs alongside molecules, captures how molecules are made in the lab. We showcase how the component can be mixed and matched in different paradigms, such as WAEs and RL, demonstrating competitive performance on various benchmarks.

## Broader Impact

Molecular de novo design, the ability to faster discover new advanced materials, could be an important tool in addressing many present societal challenges, such as global health and climate change. For example, it could contribute towards a successful transition to clean energy, through the development of new materials for energy production (e.g. organic photovoltaics) and storage (e.g. flow batteries). We hope that our methods, by producing synthesizable molecules upfront, are a contribution to the research in this direction.

An application area for molecule de novo design we are particularly enthusiastic about and believe could lead to large positive societal outcomes is drug design. By augmenting the capabilities of researchers in this area we can reduce the cost of discovering new drugs for treating diseases. This may be particularly helpful in the development of treatments for neglected tropical diseases or orphan diseases, in which currently there are often poorer economic incentives for developing drugs.

Whilst we are excited by these positive benefits that faster molecule design could bring, it is also important to be mindful of possible risks. We can group these risks into two categories, (i) use of the technology for negative *downstream applications* (for instance if our model was used in the design of new chemical weapons), and (ii) negative *side effects* of the technology (for instance including the general downsides of increased automation, such as an increased chance of accidents). If not done carefully, increased automation can have a negative impact on jobs, and in our particular case may lead to a reduction in the control and understanding of the molecular design process. We hope that this can be mitigated by communicating clearly about near-term capabilities, using and further developing sensible benchmarks/metrics, as well as the development of tools to better explain the ML decision making process.

## Acknowledgments and Disclosure of Funding

This work was supported by The Alan Turing Institute under the EPSRC grant EP/N510129/1. We are also grateful to The Alan Turing Institute for providing compute resources. JB also acknowledges support from an EPSRC studentship. We are thankful to Gregor Simm and Bernhard Schölkopf for helpful discussions.

## Footnotes

[1]Concurrent to the present work, Gottipati et al. [24] and Horwood and Noutahi [30] propose to optimize molecules via multi-step synthesis using reinforcement learning (RL). However, both approaches are limited to linear synthesis trees, where intermediates are only reacted with a fixed set of starting materials, instead of also reacting them with other intermediates. We believe these works, which focus more on improving the underlying RL algorithms, are complementary to the model presented here, and we envision combining them in future work.

[2]We represent synthesis routes as *DAGs* (directed acyclic graphs) such that there is a one-to-one mapping between each molecule and each node, even if it occurs in multiple reactions (see the Appendix for an example).

[3]Code is provided at `https://github.com/john-bradshaw/synthesis-dags`.

[4]We reimplemented the CVAE and GVAE models in PyTorch and found that our implementation is significantly better than [44]'s published results. We believe this is down to being able to take advantage of some of the latest techniques for training these models (for example $\beta$-annealing[1, 28]) as well as hyperparameter tuning.

[5]Note, like reaction predictors, these tools are imperfect, but still (we believe) offer a sensible current method for evaluating our models.

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
