[Supplementary Material]

# Appendix for: Barking up the right tree: an approach to search over molecule synthesis DAGs

**John Bradshaw**
University of Cambridge
MPI for Intelligent Systems
jab255@cam.ac.uk

**Brooks Paige**
University College London
The Alan Turing Institute
b.paige@ucl.ac.uk

**Matt J. Kusner**
University College London
The Alan Turing Institute
m.kusner@ucl.ac.uk

**Marwin H. S. Segler**
WWU Münster
Microsoft Research Cambridge, UK
marwin.segler@wwu.de

**José Miguel Hernández-Lobato**
University of Cambridge
The Alan Turing Institute
Microsoft Research Cambridge, UK
jmh233@cam.ac.uk

This document contains the Appendix for the paper "Barking up the right tree: an approach to search over molecule synthesis DAGs". It is broken up into several sections:

**Section A** The first section provides further examples of synthesis DAGs (directed acyclic graphs) and explains how they differ from trees.

**Section B** The second section expands upon Section 3 of the main paper by providing further details on our model including an algorithm (Alg. 1) showing in detail the generative process.

**Section C** The third section provides further details on our experimental setup. For instance it includes details of how we generate our dataset, and how the synthesis score is calculated. It also contains details of the hyperparameters considered and the computing infrastructure used.

**Section D** The fourth section provides further experimental results, such as additional sanity checks for the generation task as well as further figures and tables for the optimization tasks. We also provide details of the best molecules found when optimizing for QED and penalized logP, given the popularity of these metrics in previous work.

**Section E** The fifth and final section delves further into the related work discussed in the introduction and related work section of the main paper, giving a more detailed background on the problem that we are tackling and further context on previous approaches.

## A  Further examples of synthesis DAGs

**Remdesivir**  We show a simplified synthesis DAG (directed acyclic graph) of Remdesivir [1] in Figure 1. This demonstrates a case where two intermediate products need to be created as reactants for a reaction, and so differs from some of the other simpler synthesis DAGs presented in our work, which consist of the sequential addition of simple building blocks.

**Synthetic routes as DAGs**  Figure 2 shows the difference between representing synthetic routes as a DAG (directed acyclic graph), as we do in the main paper, and as a tree. In the DAG formalism there is a one-to-one mapping between each molecule and each node (see for example $Br_2$ in the aforementioned figure). For complete synthetic routes, it is easy to convert from one formalism to another by duplicating or folding nodes into each other.

The advantage of the DAG formalism comes at generation time. During generation, when using the DAG formalism, we only need to generate each unique molecule once. This can be advantageous if a

Figure 1: Simplified synthesis DAG for remdesivir (note ignoring some reagents, conditions and details of chirality) [1]. The remdesivir DAG demonstrates a case where two intermediates products need to be formed to then act later as reactants ("Bn" denotes benzyl).

complex intermediate product, which requires several steps to create, needs to be used multiple times in the graph, as in the DAG formalism it only needs to be created once.

(a) representing using *DAG formalism*          (b) representing using *Tree formalism*

Figure 2: Difference between representing the synthesis route as a DAG (with one-to-one mapping between each unique molecule and each node) or as a tree (a special form of directed acyclic graph where each node can only have one successor). Although the representations can easily be converted from one form to another, when generating, if a complex intermediate needs to be reused, then it would have to be generated twice in the tree formalism.

# B  Further details about our model

In this section we provide further details of our model. Our explanation is further broken down into three subsections. In the first we provide more details on our generative model for synthesis DAGs, including pseudocode for the full generative process. In the second subsection we provide further details on how we use a reaction predictor (here: a Molecular Transformer) for reaction prediction to fill in the products of reactions at test time. In the third and final subsection we provide further information on the fine-tuning setup. The description of the hyperparameters and specific architectures used in our models are given in the next section and further details can also be found by referring to our code.

## B.1  A generative model of synthesis DAGs

In this subsection we provide a more thorough description of our generative model for synthesis DAGs. We first recap the notation that we use in the main paper. We formally represent the DAG, $\mathcal{M}$, as a sequence of actions, with $\mathcal{M} = [V^1, \ldots, V^L]$. Alongside this we denote the associated action types as $\boldsymbol{A} = [A^1, \ldots, A^L]$. The action type entries $A^l$ take values in $\{A1, A2, A3\}$, corresponding to the three action types. The action type entry at a particular step, $A^l$, is fully defined by the actions (and action types) chosen previously to this time $l$, the exact details of which we shall come back to later. Finally the set of molecules existing in the DAG at time $l$ are denoted (in an abuse of our notation) by $M_{<l}$.

**Actions and the values that they can take**  We now describe the potential values that the actions can take. These depend on the action type at the step, and we denote this conditioning as $V^l_{|A^l}$. For example for node addition actions $A^l = A1$, the possible values of $V^l$ (i.e. $V^l_{|A_l=A1}$) are either 'B' for creating a new building block node, or 'P' for a new product node. Building-block actions $A^l = A2$

have corresponding values $V^l \in \mathcal{R}$, which determine which building block becomes a new 'leaf' node in the DAG. Connectivity choice actions $A^l = $ A3 have values $V^l \in M_{<l} \cup \{\nrightarrow_{\mathcal{I}}, \nrightarrow_{\mathcal{F}}\}$, where $M_{<l}$ denotes the current set of all molecules present in the DAG; selecting one of these molecules adds an edge into the new product node. The symbol $\nrightarrow_{\mathcal{I}}$ is an intermediate product stop symbol, indicating that the new product node has been connected to all its reactants (i.e. an intermediate product has been formed); the symbol $\nrightarrow_{\mathcal{F}}$ is a final stop symbol, which triggers production of the final product and the completion of the generative process.

As hinted at earlier, the action type, $A^l$ is defined by the previous actions $V^1, \ldots, V^{l-1}$ and action types (see also Figure 2 in the main paper). More specifically, this happens as follows:

$V^{l-1} = $ 'B', then the next action type is building block selection, $A^l = $ A2.

$A^{l-1} = $ A2 , then the next action type is again node addition, $A^l = $ A1 (as you will have selected a building block on the previous step).

$V^{l-1} = $ 'P', then the next action type is connectivity choice, $A^l = $ A3, to work out what to connect up to the product node previously selected.

$A^{l-1} = $ A3 then:

- if $V^{l-1} = \nrightarrow_{\mathcal{I}}$ then the next action type is to choose a new node again, i.e. $A^l = $ A1;
- if $V^{l-1} = \nrightarrow_{\mathcal{F}}$ the generation is finished;
- if $V^{l-1} \in M_{<l}$ then connectivity choice continues, i.e. $A^l = $ A3.

**Our generative process over these actions** Our model is shown at a high level in Figure 3 (see also Figure 3 of the main paper), which serves to provide an intuitive understanding of the generative process. The overall structure of the probabilistic model is rather complex, as it depends on a series of branching conditions: we therefore give pseudocode for the entire generative procedure in detail as a probabilistic program in Algorithm 1. The program described in Alg. 1 defines a distribution over DAG serializations; running it forward will sample from the generative process, but it can equally well be used to evaluate the probability of a DAG $\mathcal{M}$ of interest by instead accumulating the log probability of the sequence at each distribution encountered during the execution of the program. Note that given our assumption that all reactions are deterministic and produce a single primary product, product molecules do not appear in our decomposition.

## B.2 Reaction prediction

As described in the main paper we use the Molecular Transformer [2] for reaction prediction. We use pre-trained weights (trained on a processed USPTO [3, 4] dataset without reagents). Furthermore, we treat the transformer as a black box oracle and so make no further adjustments to these weights when training our model. Therefore, please note that our algorithm is not restricted to only using the Molecular Transformer — any reaction prediction model can be employed with our algorithm, and our algorithm will benefit from the development of stronger reaction predictors in the future. We take the top one prediction from the transformer as the prediction for the product, and if this is not a valid molecule (determined by RDKit) then we instead pick one of the reactants randomly.

When running our model at prediction time there is the possibility of getting loops (and so no longer predicting a DAG) if the output of a reaction (either intermediate or final) creates a molecule which already exists (in the DAG) as a predecessor of one of the reactants. A principled approach one could use to deal with this when using a probabilistic reaction predictor model, such as the Molecular Transformer, is to mask out the prediction of reactions that cause loops in the reaction predictor's beam search. However, in our experiments we want to keep the reaction predictor as a black box oracle, for which we send reactants and for which it sends us back a product. Therefore, to deal with any prediction-time loops we go back through the DAG, before and after predicting the final product node, and remove any loops we have created by choosing the first path that was predicted to each node.

## B.3 Fine-tuning

The algorithm we use for fine-tuning is given in Algorithm 2.

Figure 3: This is an expanded version of Figure 3 in the main paper showing all the actions required to produce the demonstration, example DAG for paracetamol (see also Figure 1 and 2 in the main paper). A shared RNN (recurrent neural network) provides a context vector for the different action networks. Based on this context vector, each type of action network chooses an action to take (some actions are masked out as they are not allowed, for instance suggesting a building block already in the graph, or choosing to make an intermediate product before choosing at least one reactant). Note embeddings of molecular graphs are computed using a GNN (graph neural network). The initial hidden vector of the shared RNN is initialized using a latent vector $z$ in our autoencoder model DoG-AE; in DoG-Gen it is set to a constant. The state of the DAG at each stage of the generative process is indicated in the dotted gray circles.

**Algorithm 1** Probabilistic simulator for serialized DAGs

---

**Require:** Action networks for node addition, building block molecular identity, and connectivity choice: $\mathsf{na}(\cdot)$, $\mathsf{bbmi}(\cdot)$, $\mathsf{cc}(\cdot)$;

**Require:** Reaction predictor: $\mathsf{Product}(\cdot)$;

**Require:** Context RNN: $\mathbf{c}^l = r(\mathbf{c}^{l-1}, \boldsymbol{e}^l)$;

**Require:** Continuous latent variable: $\boldsymbol{z}$

**Require:** Linear projection for mapping continuous latent to RNN initial hidden: $\mathsf{Lin}(\cdot)$.

**Require:** Gated graph neural network, $\mathsf{GGNN}(\cdot)$, for computing molecule embeddings

**Require:** Learnable embeddings for abstract actions: $\boldsymbol{h}_{\text{'B'}}$, $\boldsymbol{h}_{\text{'P'}}$, $\boldsymbol{h}_{\nrightarrow_{\mathcal{I}}}$, and $\boldsymbol{h}_{\nrightarrow_{\mathcal{F}}}$.

1: Initialize DAG $\mathcal{M} \leftarrow [V^1 = \text{'B'}]$, Initialize $\boldsymbol{A} \leftarrow [A^1 = \text{A1}, A^2 = \text{A2}]$
2: Initialize molecule set $M \leftarrow \{\}$ and set of unused reactants $U \leftarrow \{\}$ $\quad\triangleright$ Track all / all unused molecules
3: $\mathbf{c}^1 \leftarrow \mathsf{Lin}(\boldsymbol{z})$ $\quad\triangleright$ Z initializes the first hidden state of the recurrent NN
4: $\boldsymbol{e}^2 \leftarrow \boldsymbol{h}_{\text{'B'}}$ $\quad\triangleright$ Initial input into RNN reflects that new node added on first step.
5: **while** $V^{|\mathcal{M}|} \neq \; \nrightarrow_{\mathcal{F}}$ **do** $\quad\triangleright$ Loop until stop symbol
6: $\quad l \leftarrow |\mathcal{M}| + 1$
7: $\quad \mathbf{c}^l \leftarrow r(\mathbf{c}^{l-1}, \boldsymbol{e}^l)$ $\quad\triangleright$ Update context
8: $\quad$ **if** $A^l = \text{A1}$ **then** $\quad\triangleright$ Add a new node;
9: $\quad\quad \boldsymbol{w} \leftarrow \mathsf{na}(\mathbf{c}^l); \quad \boldsymbol{B} \leftarrow \text{STACK}([\boldsymbol{h}_{\text{'B'}}, \boldsymbol{h}_{\text{'P'}}])$
10: $\quad\quad \text{logits} \leftarrow \boldsymbol{w}\boldsymbol{B}^T$
11: $\quad\quad V^l \sim \text{softmax}(\text{logits})$
12: $\quad\quad$ **if** $V^l = \text{'B'}$ **then** $\quad\triangleright$ new building block
13: $\quad\quad\quad A^{l+1} \leftarrow \text{A2}; \quad \boldsymbol{e}^{l+1} \leftarrow \boldsymbol{h}_{\text{'B'}}$
14: $\quad\quad$ **else if** $V^l = \text{'P'}$ **then** $\quad\triangleright$ new product
15: $\quad\quad\quad A^{l+1} \leftarrow \text{A3}; \quad \boldsymbol{e}^{l+1} \leftarrow \boldsymbol{h}_{\text{'P'}}$
16: $\quad\quad\quad$ Initialize intermediate reactant set $R \leftarrow \{\}$ $\quad\triangleright$ Will temporarily store *working* reactants
17: $\quad\quad\quad \text{stop\_actions} \leftarrow [\boldsymbol{h}_{\nrightarrow_{\mathcal{F}}}]$ $\quad\triangleright$ You cannot stop for intermediate product until at least one reactant
18: $\quad$ **else if** $A^l = \text{A2}$ **then** $\quad\triangleright$ Pick building block molecular identity
19: $\quad\quad \boldsymbol{w} \leftarrow \mathsf{bbmi}(\mathbf{c}^l); \quad \boldsymbol{B} \leftarrow \text{STACK}([\mathsf{GGNN}(g) \text{ for } g \text{ in } \mathcal{R} \setminus M])$
20: $\quad\quad \text{logits} \leftarrow \boldsymbol{w}\boldsymbol{B}^T$
21: $\quad\quad V^l \sim \text{softmax}(\text{logits})$ $\quad\triangleright$ Pick building block molecule
22: $\quad\quad A^{l+1} \leftarrow \text{A1}; \quad \boldsymbol{e}^{l+1} \leftarrow \mathsf{GGNN}(V^l)$
23: $\quad\quad M \leftarrow M \cup \{V^l\}, U \leftarrow U \cup \{V^l\}$
24: $\quad$ **else if** $A^l = \text{A3}$ **then** $\quad\triangleright$ Connectivity choice
25: $\quad\quad \boldsymbol{w} \leftarrow \mathsf{cc}(\mathbf{c}^l); \quad \boldsymbol{B} \leftarrow \text{STACK}([\mathsf{GGNN}(g) \text{ for } g \text{ in } M \setminus R] + \text{stop\_actions})$
26: $\quad\quad \text{logits} \leftarrow \boldsymbol{w}\boldsymbol{B}^T$
27: $\quad\quad V^l \sim \text{softmax}(\text{logits})$ $\quad\triangleright$ Pick either (i) molecule to connect to, or (ii) to end and create product
28: $\quad\quad$ **if** $V^l = \nrightarrow_{\mathcal{I}}$ **then** $\quad\triangleright$ Selected an intermediate product so run the reaction
29: $\quad\quad\quad M^{\text{new}} \leftarrow \mathsf{Product}(R)$
30: $\quad\quad\quad M \leftarrow M \cup \{M^{\text{new}}\}, U \leftarrow U \cup \{M^{\text{new}}\}$
31: $\quad\quad\quad A^{l+1} \leftarrow \text{A1}; \quad \boldsymbol{e}^{l+1} \leftarrow \boldsymbol{h}_{\nrightarrow_{\mathcal{I}}}$
32: $\quad\quad$ **else if** $V^l \in M$ **then** $\quad\triangleright$ Selected an extra reactant
33: $\quad\quad\quad R \leftarrow R \cup \{V^l\}$ $\quad\triangleright$ Update reactant set
34: $\quad\quad\quad U \leftarrow U \setminus \{V^l\}$ $\quad\triangleright$ Remove from pool of "unused" molecules
35: $\quad\quad\quad A^{l+1} \leftarrow \text{A3}; \quad \boldsymbol{e}^{l+1} \leftarrow \mathsf{GGNN}(V^l)$
36: $\quad\quad\quad \text{stop\_actions} \leftarrow [\boldsymbol{h}_{\nrightarrow_{\mathcal{I}}}, \boldsymbol{h}_{\nrightarrow_{\mathcal{F}}}]$ $\quad\triangleright$ Now you can stop for both final or intermediate product
37: $\quad$ Update $\mathcal{M} \leftarrow [V^1, \dots, V^l]; \quad \boldsymbol{A} \leftarrow [A^1, \dots, A^l]$
38: Predict final product $M_T \leftarrow \mathsf{Product}(R \cup U)$ $\quad\triangleright$ The final product considers both $R$ and $U$
39: **return** $\mathcal{M}, M_T$

**Algorithm 2** Synthesis DAG Fine-Tuning. Note that for fine-tuning we use the model DoG-Gen, in which $z$ is always set at $\mathbf{0}$, hence we drop our specific dependence on $z$ in this algorithm.

---

**Require:** Initial model $p_\theta(\mathcal{M})$, iterations $I$, threshold $K$, sample size $N$, objective $h(\cdot)$, pool of seen synthesis DAGs (initially empty), $P$.

1: Compute the score $h(\cdot)$ of all products in the initial training set and add to $P$
2: **for** $i = 1, \ldots, I$ **do**
3:      Sample $N$ DAGs from $p_\theta(\mathcal{M})$
4:      Compute the score $h(\cdot)$ of the $N$ products and add to pool, $P$
5:      Select the $K$ DAGs with the highest score from $P$
6:      Run two training epochs on $\theta$ using these $K$ DAGs as training data

7: **return** updated model $p_\theta(\mathcal{M})$, pool of all seen synthesis DAGs (ranked) $P$

---

## C    Further experimental details

This section provides further details about aspects of our experiments. We start by describing how we create a dataset of synthesis DAGs for training. We then describe how the synthesis score we use in the optimization experiments is calculated. Finally, the latter subsections provide specific details on the hyperparameters we use. Further details about our experiments can also be found in our code.

### C.1    Creating a dataset of synthesis DAGs

In this subsection we describe how we create a dataset of synthesis DAGs, with a high level illustration of the process given in Figure 4. Further details can also be found by referring to our code. The creation of our synthesis DAG dataset starts by collecting the reactions from the USPTO dataset [4], using the processed and cleaned version of this dataset provided by [3, §4]. We filter out reagents (molecules that do not contribute any atoms to the final product) and multiple product reactions (97% of the dataset is already single product reactions) using the approach of Schwaller et al. [5, §3.1].

This processed reaction data is then used to create a reaction network [6–8]. To be more specific, we start from the reactant building blocks specified in Bradshaw et al. [9, §4] as initial molecule nodes in our network, and then iterate through our list of processed reactions adding any reactions (and the associated product molecules) (i) that depend only on molecule nodes that are already in our network, and (ii) where the product is not an initial building block. This process repeats until we can no longer add any of our remaining reactions.

This reaction network is then used to create one synthesis DAG for each molecule. To this end, starting from each possible (non building block) molecule node in our reaction network, we step backwards through the network until we find a sub-graph of the reaction network (without any loops) with initial nodes that are from our collection of building blocks. When there are multiple possible routes we pick one. This leaves us with a dataset of 72008 synthesis DAGs, which we use approximately 90% of as training data and split the remainder into a validation dataset (of 3601 synthesis DAGs) and test dataset (of 3599 synthesis DAGs).

The training set DAGs have an average of 4.6 nodes, with the final molecules containing an average of 20.5 heavy atoms and 21.7 bonds (between heavy atoms). The average number of actions required to construct these DAGs is 11, as each reactant often contributes several atoms and bonds to the product.

### C.2    Synthesizability Score

The synthesizability score is defined as the geometric mean of the nearest neighbor reaction similarities:

$$\sqrt[|R|]{\prod_{r \in R} \kappa(r, \text{nn}(r))} \tag{1}$$

where $R$ is the list of reactions making up a synthesis DAG, $r \in R$ are the individual reactions in the DAG, $\text{nn}(r)$ is the nearest neighbor reaction in the chemical literature in Morgan fingerprint space, and $\kappa(\cdot, \cdot)$ is Tanimoto similarity over Morgan reaction fingerprints [10].

**Input:** USPTO dataset as atom-mapped SMILES lines

```
[CH2:15]([CH:16]([CH3:17])[CH3:18])[Mg+:19].[CH2:20]1[O:21][C
H2:22][CH2:23][CH2:24]1.[Cl-:14].
[OH:1][c:2]1[n:3][cH:4][c:5]([C:6](=[O:7])[N:8]([O:9][CH3:10]
)[CH3:11])[cH:12][cH:13]1>>
[OH:1][c:2]1[n:3][cH:4][c:5]([C:6](=[O:7])[CH2:15][CH:16]([CH
3:17])[CH3:18])[cH:12][cH:13]1 15-19;6-15;6-8

[CH3:14][NH2:15].[N+:1](=[O:2])([O-
:3])[c:4]1[cH:5][c:6]([C:7](=[O:8])[OH:9])[cH:10][cH:11][c:12
]1[Cl:13].
[OH2:16]>>
[N+:1](=[O:2])([O-
:3])[c:4]1[cH:5][c:6]([C:7](=[O:8])[OH:9])[cH:10][cH:11][c:12
]1[NH:15][CH3:14] 12-13;12-15
...
```
1.

Remove atom-mappings and reagents so that we have a list of reactions, where each reaction is a mapping from a set of reactants to one product molecule.

2.

Build up reaction network by starting from specified building blocks then repeatedly iterating through the available reactions and adding any that can be made from molecules already existing in the graph.

3.

Create synthesis DAG by picking each intermediate product in turn and searching back through the reaction network until we have found a sub-graph that starts from building blocks only.

4.

Figure 4: An illustration of how we create a dataset of synthesis DAGs from a dataset of reactions. We first clean up the reaction dataset by removing reagents (molecules which do not contribute atoms to the final product) and any reactions which lead to more than one product. We then form a modified reaction network (we do not allow loops back to building block molecules), which is a directed graph showing how molecules are linked to others through reactions. This process starts by adding molecule nodes corresponding to our initial building blocks. We then repeatedly iterate through our list of reactions and gradually add reaction nodes (and their associated product nodes) to the graph if both (i) the corresponding reaction's reactants are a subset of the molecule nodes already in the graph, and (ii) the product is not a building block. Finally for each possible product node we iterate back through the directed edges until we have selected a subgraph without any loops, where the initial nodes are members of our set of building blocks.

## C.3   Atom features used in DoG models

The atom features we use as input to our graph neural networks (GNNs) operating on molecules are given in Table 1. These features are chosen as they are used in Gilmer et al. [11, Table 1] (we make the addition of an expanded one-hot atom type feature, to cover the greater range of elements present in our molecules).

Table 1: Atom features we use as input to the GGNN. These are calculated using RDKit.

| Feature | Description |
|---|---|
| Atom type | 72 possible elements in total, one hot |
| Atomic number | integer |
| Acceptor | boolean (accepts electrons) |
| Donor | boolean (donates electrons) |
| Hybridization | One hot (SP, SP2, SP3) |
| Part of an aromatic ring | boolean |
| H count | integer |

## C.4 Implementation details for DoG-AE

In this subsection we describe specifics of our DoG-AE model used to produce the results in Table 1 of the main paper.

**Forming molecule embeddings** For forming molecule embeddings we use a GGNN (Gated Graph Neural Network) [12]; this operates on the atom features described in Table 1. This graph neural network (GNN) was run for 4 propagation steps to update the node embeddings, before these embeddings were projected down to a 50 dimensional space using a learnt linear projection. The node embeddings were then combined to form molecule embeddings through a weighted sum. The same GNN architecture was shared between the encoder and the decoder.

**Encoder** The encoder consists of two GGNNs. The first, described above, creates molecule embeddings which are then used to initialize the node embeddings in the synthesis DAG. The synthesis DAG node embeddings, which are 50 dimensional, are further updated using a second GGNN. Using the notation introduced in Section 3.2.1 of the main paper, we have:

$$g_e(\mathbf{m}_M^t, \mathbf{e}_M^{t-1}) = \text{GRU}(\mathbf{m}_M^t, \mathbf{e}_M^{t-1}) \tag{2}$$

$$g_m(\mathbf{e}_M^{t-1}, \mathbf{e}_{N(M)}^{t-1}) = \sum_{j \in N(M)} \text{Lin}_1(\mathbf{e}_j^{t-1}) \tag{3}$$

where $\text{GRU}(\cdot)$ is a gated recurrent unit (GRU) [13] and $\text{Lin}_1(\cdot)$ is a learnt linear projection. Seven propagation steps of message passing are carried out on the DAG (i.e. $T' = 7$), and the messages are passed forward on the DAG from the 'leaf' nodes to the final product node. Finally, the node embedding of the final product molecule node in the DAG is passed through an additional linear projection to parameterize the mean and log variance of independent Gaussian distributions over each dimension of the latent variable, $\mathbf{z}$, i.e.:

$$g_\mu(\mathbf{E}_{\mathcal{M}}^{T'}) = \text{Lin}_2\left([\mathbf{E}_{\mathcal{M}}^{T'}]_{M^T}\right) \tag{4}$$

$$g_{\log \sigma^2}(\mathbf{E}_{\mathcal{M}}^{T'}) = \text{Lin}_3\left([\mathbf{E}_{\mathcal{M}}^{T'}]_{M^T}\right) \tag{5}$$

where we are using the notation $[\mathbf{E}_{\mathcal{M}}^{T'}]_{M^T}$ to indicate the indexing of the node embedding corresponding to the final product node ($M^T$) in the DAG ($\mathcal{M}$), after $T'$ stages of message passing.

**Decoder** For the decoder we use a 3 layer GRU RNN [13] to compute the context vector. The hidden layers have a dimension of 200 and whilst training we use a dropout rate of 0.1. For initializing the hidden layers of the RNN we use a linear projection (the parameters of which we learn) of $z$. The action networks are feedforward neural networks with one hidden layer (dimension 28) and ReLU activation functions. For the abstract actions (such as 'B'or 'P') we learn 50 dimensional embeddings, such that these embeddings have the same dimensionality as the molecule embeddings we compute.

**Training** We train our model, with a 25 dimensional latent space, using the Adam optimizer [14], an initial learning rate of 0.001, and a batch size of 64. We train the autoencoder using the Wasserstein autoencoder loss [15] (Eq. 2 main paper), with $\lambda = 10$ and an inverse multiquadratics kernel for computing the MMD-based penalty, as this is what is used in Tolstikhin et al. [15, §4].

Our model, DoG-AE, is trained using teacher forcing for 400 epochs (each epoch took approximately 7 minutes) and we multiplied the learning rate by a factor of 0.1 after 300 and 350 epochs. DoG-AE obtains a reconstruction accuracy (on our held out test set) of 65% when greedily decoding (greedy in the sense of picking the most probable action at each stage of decoding). If we try instead decoding by sampling 100 times from our model and then sorting based on probability we obtain a slightly improved reconstruction accuracy of 66%.

Initially in our experiments we trained for a shorter period, but we increased the training time after monitoring the reconstruction rate on our held out validation dataset. Likewise for debugging purposes we initially trained with a smaller architecture. To be more specific, we tried a two layer GRU RNN in our decoder with 28 dimensional hidden layers. We also set the molecule embedding sizes to be 28 dimensional. We increased the size of our architecture after we found that it improved results. However, we have yet to attempt a thorough hyperparameter search which may improve results further.

**Timings**  Using a NVIDIA K80 GPU it takes $\approx 7$ mins to run a training epoch for DoG-AE ($\approx$ 0.4 secs per batch of 64). At inference time, where we do not initially have access to the complete sequence, we usually run larger batches, due to the fixed costs and latency of communicating with a Molecular Transformer server. It takes $\approx 29$ secs to carry out *per batch of 200 DAGs* (and $\approx 12$ secs per batch of 64 DAGs). Our approach could be further sped up by using faster reaction predictors.

## C.5  Implementation details for DoG-Gen

For DoG-Gen we also used a GGNN to create molecule embeddings in a similar way to DoG-AE. The GGNN was run for 5 rounds of message passing to form 80 dimensional node embeddings; these node embeddings were aggregated into a 160 dimensional molecule embedding through a linear projection and weighted sum. For generating the context vector we use a 3 layer GRU RNN with 512 dimensional hidden layers. The action networks used were feed-forward neural networks with one hidden layer of dimension 28 and ReLU activation functions. We trained our model for 30 epochs, as we found using our validation dataset, that training for longer would lead to overfitting. Aside from monitoring the training time we have not tried tuning other hyperparameters, which we believe could lead to better results.

For optimization we start by evaluating the score on every synthesis DAG in our training and validation datasets; we then run 30 stages of fine-tuning, sampling 7000 synthesis DAGs at each stage and updating the weights of our model using the best 1500 DAGs seen at that point as a fine-tuning dataset. For both our model and the baselines, when reporting the GuacaMol benchmark property score we report the score obtained by the best individual molecule in the group.

To parallelize over the GuacaMol optimization tasks we used Jug [16].

## C.6  Details of baselines for generation tasks

We used the following implementations for the baselines:

- SMILES LSTM [17]: `https://github.com/benevolentAI/guacamol_baselines`.
- JT-VAE [18]: `https://github.com/wengong-jin/icml18-jtnn` (we used the updated version of their code, ie the `fast_jtnn` version)
- CGVAE [19]: `https://github.com/Microsoft/constrained-graph-variational-autoencoder`
- Molecule Chef [9]: `https://github.com/john-bradshaw/molecule-chef`

For the CVAE, GVAE and GraphVAE baselines we used our own implementations. We tuned the hyperparameters of these models on the ZINC or QM9 datasets so that we were able to get at least similar (and often better) results compared to those originally reported in Kusner et al. [20], Simonovsky and Komodakis [21].

When training the GraphVAE on our datasets we exclude any molecules with greater than 20 heavy atoms, as this procedure was found in the original paper to give better performance when training on ZINC [21, §4.3]. We use a 40 dimensional latent space, a GGNN [12] for the encoder, and use max-pooling graph matching during training.

For the CVAE and GVAE we use 72 dimensional latent spaces. We multiply the KL term in the VAE loss by a parameter $\beta$ [22, 23]; this $\beta$ term is then gradually annealed in during training until it reaches a final value of $0.3$. We use a 3 layer GRU RNN [13] for the decoder with 384 dimensional hidden layers. The encoder is a 3 layer bidirectional GRU RNN also with 384 dimensional hidden layers.

## C.7  Computing infrastructure used

We mostly used a NVIDIA Tesla K80 GPU when training and sampling from our models (predominately using NC6 and NC12 virtual machines on Azure). At sampling time we ran a Molecular Transformer in server mode on a separate K80 GPU.

For training the baselines for the generation task we used a mixture of NVIDIA Tesla K80, GeForce GTX 1080 Ti and P100 GPUs. The NVIDIA P100 GPU was used for training the CGVAE as this model required a GPU with a large memory.

# D Further experimental results

## D.1 Generation – Additional Sanity Checks

Following prior work [24, 25], we also plot the distributions of the Quantitative Estimate of Drug Likeness score (QED) [26], Synthetic Accessibility score (SA) [27], and the octanol-water partition coefficient (logP) [28] over the generated molecules. The results of this are shown in Figure 5. Apart from the GraphVAE and CGVAE, all models closely match the training set. We again see this as a useful model sanity check.

Figure 5: KDE (kernel density estimation) plots for the distribution of drug likeness score (QED) [26], synthetic accessibility score (SA) [27], and the octanol-water partition coefficient (logP) for 20k molecules sampled from each of the various models, compared to the training set. Plot done in same style as [24, Fig.3]

## D.2 Optimization

**Further plots and tables for the GuacaMol optimization tasks** We provide further figures and tables for the GuacaMol optimization tasks. Figure 6 shows the scores of the best molecules found on each of the tasks by the various methods, particularly distinguishing between the cases in which (a) no synthetic route can be found, (b) a synthetic route can be found, and (c) a synthetic route can be found *and* the found route has a synthesis score over 0.9. Figure 7 and Table 2 show the fraction of the top 100 molecules proposed by each method for each task for which a synthetic route can be found. Table 3 shows the average synthesis score over the 100 best molecules proposed by each method for each task. Table 4 shows the median number of synthesis steps over the top 100 best molecules for each method and task, where the median has been calculated considering only those molecules in the top 100 for which a synthetic route can be found.

As an additional baseline, we reimplemented the SYNOPSIS algorithm, one of the first reaction-driven de novo design algorithms based on discrete optimization (simulated annealing), proposed by Vinkers et al. [29] in 2003. The algorithm proceeds by sampling molecules from a pool (which is initially populated by all building blocks and training molecules that the other models also have access to). Then, at each iteration three molecules $m_i$ are sampled according to a Boltzmann distribution $p(m_i) = \frac{\exp(-(V_{\max}-V_{m_i})/T)}{\sum_j \exp(-(V_{\max}-V_{m_j})/T)}$, with molecules $m_i$, the score of a molecule $V_{m_i}$, the score of the currest best molecule collected so far $V_{\max}$, and a quasi-temperature $T$. For these molecules, a reaction scheme is randomly sampled, using a union of the reaction schemes published in [30, 31]. If the reaction requires a reaction partner, up to 64 building blocks are uniformly-randomly sampled from all matching building blocks. The resulting molecules are scored, and added back to the pool and to an output collection, from which the results are eventually returned. The algorithm is run for 1000 iterations, generating $\approx 100,000$ molecules per task. Table 5 shows the best molecule scores on the GuacaMol tasks found by this reimplementation of Vinkers et al. [29]'s algorithm in comparison to the DoG-Gen algorithm. The results indicate that DoG-Gen also performs favorably in comparison to an established reaction-driven de novo design algorithm.

Figure 6: The score of the best molecule found by the different approaches over a series of ten GuacaMol tasks [32, §3.2]. Scores range between 0 and 1, with 1 being the best. We also differentiate between the synthesizability of the different best molecules found by the hatching on the bar, differentiating between three different cases: (1) the best molecule found (regardless of synthesizability) is shown with a circular hatched bar, (2) the best molecule found with a synthetic route with the diagonally hatched bar, and (3) the best molecule found which has a synthetic score greater than or equal to 0.9 is given a solid bar. Note that later bars occlude previous bars. The dotted gray line for each task represents the score one obtains if picking the best molecule in the training set.

Figure 7: The fraction of the top 100 molecules proposed that for which a synthetic route can be found, over a series of ten Guacamol tasks [32, §3.2].

Figure 8: The mean of the synthesis score of the top 100 molecules proposed by each method, over a series of ten GuacaMol tasks [32, §3.2].

Table 2: Fraction of the top 100 molecules suggested by each method for which a synthetic route can be found.

|  | Graph GA | SMILES LSTM | SMILES GA | DoG-Gen |
|---|---|---|---|---|
| Amlodipine | 0.52 | 0.13 | 0.00 | 1.00 |
| Aripiprazole | 0.98 | 0.72 | 0.29 | 0.99 |
| Deco Hop | 0.09 | 0.05 | 0.16 | 0.95 |
| Osimertinib | 0.27 | 0.09 | 0.00 | 0.67 |
| Perindopril | 0.25 | 0.15 | 0.02 | 0.82 |
| Ranolazine | 0.00 | 0.77 | 0.00 | 0.63 |
| Scaffold Hop | 0.54 | 0.02 | 0.68 | 0.91 |
| Sitagliptin | 0.00 | 0.96 | 0.50 | 0.98 |
| Valsartan | 0.85 | 0.87 | 0.89 | 1.00 |
| Zaleplon | 0.72 | 1.00 | 0.53 | 1.00 |

Table 3: Average synthetic score of the top 100 molecules suggested by each method.

|  | Graph GA | SMILES LSTM | SMILES GA | DoG-Gen |
|---|---|---|---|---|
| Amlodipine | 0.35 | 0.09 | 0.00 | 0.86 |
| Aripiprazole | 0.91 | 0.62 | 0.24 | 0.98 |
| Deco Hop | 0.07 | 0.04 | 0.12 | 0.74 |
| Osimertinib | 0.20 | 0.06 | 0.00 | 0.57 |
| Perindopril | 0.19 | 0.09 | 0.02 | 0.68 |
| Ranolazine | 0.00 | 0.69 | 0.00 | 0.55 |
| Scaffold Hop | 0.36 | 0.02 | 0.52 | 0.71 |
| Sitagliptin | 0.00 | 0.62 | 0.42 | 0.83 |
| Valsartan | 0.72 | 0.79 | 0.82 | 0.88 |
| Zaleplon | 0.48 | 0.87 | 0.48 | 0.82 |

Table 4: Table showing the median number of synthetic steps for each molecule found in the top 100 (median calculated only over synthesizable molecules). Hyphens ('-') indicate no synthesizable molecule was suggested in the top 100 by that method on that task.

|  | Graph GA | SMILES LSTM | SMILES GA | DoG-Gen |
|---|---|---|---|---|
| Amlodipine | 7.0 | 6.0 | - | 3.5 |
| Aripiprazole | 3.0 | 6.0 | 5.0 | 2.0 |
| Deco Hop | 5.0 | 8.0 | 3.5 | 5.0 |
| Osimertinib | 8.0 | 9.0 | - | 7.0 |
| Perindopril | 7.0 | 11.0 | 6.0 | 7.0 |
| Ranolazine | - | 7.0 | - | 8.0 |
| Scaffold Hop | 7.0 | 8.0 | 5.0 | 4.0 |
| Sitagliptin | - | 7.0 | 3.0 | 3.0 |
| Valsartan | 6.0 | 3.0 | 2.0 | 3.0 |
| Zaleplon | 6.0 | 3.0 | 4.0 | 3.0 |

Table 5: Score of the best molecule found by DoG-Gen and the SYNOPSIS [29] inspired, discrete optimization algorithm on the GuacaMol tasks

|  | DoG-Gen | SYNOPSIS |
|---|---|---|
| Amlodipine | 0.80 | 0.63 |
| Aripiprazole | 1.00 | 0.87 |
| Deco Hop | 1.00 | 0.88 |
| Osimertinib | 0.89 | 0.84 |
| Perindopril | 0.70 | 0.55 |
| Ranolazine | 0.87 | 0.83 |
| Scaffold Hop | 0.67 | 0.54 |
| Sitagliptin | 0.55 | 0.43 |
| Valsartan | 1.00 | 0.00 |
| Zaleplon | 0.62 | 0.52 |

**QED and penalized logP Optimization**  We also considered optimizing for QED [26] and penalized logP[1], as these metrics have been used in previous work [18, 20, 33]. We did not include this evaluation in the main paper, as similar to Brown et al. [32, §5.2 & §8.5] we found that our approach was able to obtain very high scoring molecules quite easily, which meant that this score was less useful in investigating the advantages and disadvantages of each method. However, for reference we note that DoG-Gen found a molecule with top QED score of 0.948[2] and a top penalized logP score of 124.8[3] (we run the penalized logP optimization for only 15 rounds of fine-tuning as it quickly exploits the objective by producing a molecule with a long carbon chain.).

## D.3 Retrosynthesis

Our generative model of synthesis DAGs can be seen as a parametrizable mapping from a vector of real numbers to a synthesis DAG. As a module it can be mixed and matched in different ML frameworks as we have already seen in the main paper with DoG-AE and DoG-Gen. In this section we describe some preliminary results with a third model architecture, called RetroDoG. This model consists of the composition of a GNN followed by our generative synthesis DAG model to produce a learnable mapping from a molecular graph to a synthesis DAG. By training this model on pairs of product molecules and their associated synthesis DAGs we can use this model to perform retrosynthesis (i.e. predict how a particular product can be made). While automated retrosynthesis is canonically performed using planning algorithms [34], the model described in this section would additionally allow one to feed in a potentially hard or impossible to synthesize molecule, and obtain a similar molecule which is *easy to synthesize*, which is impossible with current planners.

In order to qualitatively assess such a model we train RetroDoG on the same DAG dataset described in Section C.1. At test time we sample 200 DAGs from our model and then sort them based on Tanimoto similarity (with Morgan fingerprints) between the final product molecule and the original molecule fed into RetroDoG, before picking the best one (note this uses only the input data and not the true target data). In Figures 9, 10 and 11 we show the result of this on three molecules taken from the WHO Essential Medicines [35] list. Although we do not find a route to the exact molecule of interest, we often decode to similar final product molecules. We find that RetroDoG finds molecules that are often as similar (measured again using the Tanimoto similarity between the fingerprints) as the best in our original training dataset.

We should point out that we do not expect this preliminary approach to currently be competitive with complex synthesis planning tools such as the one proposed by Segler et al. [34]. These complex tools work in a top-down manner assessing many different routes. RetroDoG in contrast tries to construct the DAG in a bottom-up manner and is not able to roll back and adjust already made choices based on new data. Having said that, we believe such an approach as RetroDoG may be worthy of future research interest, and may for instance be useful in combination with more complex tools to amortize and reduce the cost of searching for synthetic routes.

CCCCCCCCCCCCCCCCCCCCCCCCCCCCCCCCCCCCCCCCCCCCCCCCCCCCCCCCCCCCCCCCCCCCCCCCCCCCCCCCCCCCCCCCCCC
CCCCCCCCCCCCCCCCCCCCCCCCCCCCCCCCCCCCCCCCCCCCCCCCCCCCCCCCCCCCCCCCCCCCCCCCCCCCCCCCCCCCCCCCCCC
CCCCCCCCCCCCCCCCCCCCCCCCCCCCCCCCCCCCCCCCCCCCCCCCCCCCCCCCCCCCCCCCCCCCCCCCCCCCCCCCCCCCCCCCCCC
CCCCCCCCCCCCCCCCCCCCCCCCCCCCCCCCCCCCCCCCCCCCCCCCCCCCCCCCCCCCCCCCCCCCCCCCCCCC[S+](CCCCCCCCCCCCCCC
CC)CCCCCCCCCCCCCCCCCCCCCCCCCCCCCC`

Figure 9: RetroDoG suggested DAG for ribavirin.

Figure 10: RetroDoG suggested DAG for methotrexate.

Figure 11: RetroDoG suggested DAG for epinephrine.

# E Further Background

This section contains some further background and context on molecule design, complimenting the introduction and related work sections in the main paper.

**Virtual screening and de novo design.** In computer-aided molecular discovery there are two main approaches to come up with the next molecules to make and test, virtual screening and de novo design [36]. In virtual screening (VS), large molecule libraries are pre-generated and then scored or searched on demand , sometimes using ML models [37], requiring $\mathcal{O}(N)$ time and memory/storage. However, given the combinatorial nature of molecules, which results in search spaces of the order $10^{24} - 10^{60}$, complete coverage of chemical space is neither feasible nor desirable [38–43]. Instead, the technique of *de novo design* can be used, where a molecule construction algorithm is coupled with an optimization or search procedure, and novel molecules are generated on the fly [36, 44].

When comparing different molecule generation algorithms, there is always a tradeoff between the coverage of chemical space, and synthesizability, stability, and even to some extent the sensibleness of suggested structures. Synthesizability, stability and the sensibleness depend on the application area, are difficult to define, and can all appear subtle to non-chemists. For example, although we can encode a simple notion of syntactic correctness through valency constraints (for instance having maximally four substituents at a carbon atom), these do not guarantee that a molecule is semantically valid. Semantic validity is not trivial to define, and can be related to stability or the presence of undesirable substructures. For instance, a molecule consisting of a chain of 10 nitrogen atoms with alternating single and double bonds would satisfy valency constraints, but would unlikely be stable; it would decompose or explode immediately!

We can view VS as being on one end of this tradeoff between chemical coverage and synthesizability/stability. By using pre-defined virtual libraries, we are limited in chemical coverage but can ensure that all the molecules included are sensible and synthesizable. This for instance is sometimes done by generating the library by iterating over a set of readily available starting materials, and combining them using virtual reaction schemes [30].

Inhabiting the other end of this tradeoff, would be a completely unconstrained de novo design algorithm, allowing any connecting bond between any atoms to construct the molecular graph. Although such an algorithm covers every conceivable molecule, it would generate structures not even meeting valency constraints. This can be remedied by the hand-coding of expert rules, e.g. [45]. However, arguably a more scalable approach was the development of algorithms that constructed molecules through combining together larger, sensible fragments [46] and later algorithms that constructed molecules iteratively by combining together building blocks via virtual reaction schemes[29, 47]. However, even with these approaches there is still the problem of how best to search through this action space, a difficult discrete optimization problem, and so these algorithms often resorted to optimizing greedily one-step at a time or using genetic algorithms.

**ML design-and-search techniques for molecules** A paradigm shift in molecule construction algorithms was achieved by the introduction of neural generative models, in particular variational autoencoders on sequences [48], and RNN-based language models [17, 49]. To perform optimization, autoencoder-based models map from a continuous latent space to the discrete molecule space, allowing optimization to be performed in the continuous space, for instance using Bayesian optimization, leveraging local smoothness properties and gradient information. Autoregressive-style models can be used for optimization using reinforcement learning [17, 33, 49, 50]. Some of these SMILES-based LSTMs have also been successfully used to prospectively suggest new molecules for lab-based experiments [51–53].

Even though pre-training on large molecule datasets of known molecules mitigates the issue of generating unstable and unsynthesizable molecules to a considerable extent, candidate molecules generated by current machine learning systems can still be difficult to synthesize and sometimes contain unrealistic moieties [32, 36, 54]. This is particularly the case when optimizing. So again the question becomes how can we build these constraints into our models?

**Going further than molecular string based representations** The work of Gómez-Bombarelli et al. [48] relied on string representations of molecular graphs (SMILES [55]), which although flexible can lead to invalid molecules. Extensions of this approach have therefore investigated representing

molecules with grammars [20, 56], and explicit constrained graph representations or molecular fragments [18, 19, 57–60]. However, although these particular extensions improve the validity of generated molecules, they do not explicitly consider synthesizability.

Generative ML models which construct molecules from building blocks via chemical reactions (e.g. [9, 61–63] and the work presented here) can fix this problem by design. The constraint of requiring a viable synthesis route offers a powerful form of regularization, discouraging unstable structures. Furthermore, it provides interpretability to chemists as well as a possible head start in making, and in turn testing, any proposed molecules.

## Footnotes

[1]Using the ZINC-normalized definition of this score, as computed for instance in the code of You et al. [33] at `https://github.com/bowenliu16/rl_graph_generation/blob/master/gym-molecule/gym_molecule/envs/molecule.py`.

[2]With SMILES: `Cc1cc(F)ccc1NS(=O)(=O)c1ccc2c(c1)CCO2`

[3]With SMILES: `CCCCCCCCCCCCCCCCCCCCCCCCCCCCCCCCCCCCCCCCCCCCCCCCCCCCCCCCCCCCCCCCCCCCCCCCCCCCCCCCCCCCCC