[Reviews · NeurIPS 2020]

Review 1

Summary and Contributions: This submission describes an autoencoder for directed acyclic graphs that can be applied to molecular synthesis graphs. Synthesizability is an underappreciated aspect of molecular generation. The decoder ensures that valid synthetic pathways are generated, subject to an external oracle that predicts reaction products from reactants. An optimization setting is also demonstrated where reinforcement learning is used to bias generation, neglecting the latent encoding.

Strengths: The complex process of generating a DAG is cleverly posed as a sequential decision-making process while imposing some constraints on the action space (e.g., selecting commercially-available molecules, forbidding cyclic pathways) informed by the domain. Remarkably, this actually works as demonstrated through Figure 5’s walk in the latent space. Empirical results for goal-directed optimization are thorough and convincing.

Weaknesses: I do not see any weaknesses with the current work, only exciting potential extensions. I can tell that the authors were constrained by the page limits, but the Appendix contains more complete explanations of the work.

Correctness: Yes, I have no concerns.

Clarity: The only suggestion I would make is to add a caveat to the main text that both the Molecular Transformer and the CASP oracle are imperfect; computer scientists unfamiliar with chemistry might not otherwise appreciate that point. Minor point: the (lack of) capitalization of article titles in the references is odd. Some typos include “WAE”, a missing space in Figure 5’s caption.

Relation to Prior Work: Yes.

Reproducibility: Yes

Additional Feedback: I would highly recommend discussing the reconstruction accuracy of DoG-AE (Appendix L168) in the main text. I know it’s hard to contextualize whether 65% is good or bad (and there are no other approaches to benchmark against), but this is an important statistic to mention. Further, because this is an autoregressive model, it would be appropriate to examine reconstruction accuracy on the test set when using a beam search of some kind. --- I've read the rebuttal - still strongly support acceptance


Review 2

Summary and Contributions: The submission presents a molecular generation framework that incorporates synthesizability information by directly generating chemical reaction pathways (more specifically reaction DAGs) that specify how the molecule is made. The model is shown to have competitive performance on a variety of molecule generation and optimization benchmark metrics

Strengths: Quite an interesting problem, and improvements in molecular property optimization could have significant impacts on drug discovery, which could be very relevant in the current environment The proposed architecture to generate synthesis DAGs is very interesting and allows the possibility of having multi-step non-linear synthesis routes A wide range of molecular generation baselines are benchmarked The proposed model has competitive performance on a variety of benchmark metrics Code is provided

Weaknesses: The model assumes that there exists a perfect forward reaction predictor oracle for the generation process. Given that this cannot be true, it would have been interesting to see how sensitive the proposed model is to varying error rates of the reaction predictor module, or at least some comment about this issue. Eg does the reaction DAG generation process just break if a spurious reaction prediction is made?

Correctness: Seems to be correct

Clarity: Paper is well written and arranged in a clear way. The fonts and structures in the figures are too small, eg Figure 3, 4, 5

Relation to Prior Work: Related work section is quite comprehensive

Reproducibility: Yes

Additional Feedback: The model requires a large reaction data set in order to generate sufficient quantities of synthesis DAGs. From the largest public reaction dataset (USPTO), only 72008 reaction DAGs could be extracted. This seems like quite a small training dataset. Do you think it is a significant constraining factor? Once again, the font size in the figures is way too small and hard to read, especially in print ## After rebuttal and reviewer discussion Still support acceptance of paper


Review 3

Summary and Contributions: The authors proposed a multi-step molecular synthesis route generation method, given a target molecule as input. Specifically, by representing the synthesis route as a directed acyclic graphs (DAGs), the authors proposed a hierarchical neural message passing procedure that exchanges information not only among atoms in molecules, but also among the nodes in the DAG, and develop an efficient serialization procedure for DAGs.

Strengths: (1) The paper aims to tackle a very interesting problem: reverse-engineering the synthesis procedure to generate the module of interest (2) The paper is very well-written, and the figures are very illustrative

Weaknesses: (1) Each DAG may correspond to multiple topological sorting paths; it is unclear if they are all valid synthesis procedure. If so, are they all enumerated in the training step. (2) It is unclear if all partial synthesis procedure (i.e., a subset of actions up to some moment) also corresponds to a valid and non-out-of-distribution action embedding.

Correctness: It seems reasonable, though I must admit it’s out of my expertise domain.

Clarity: Yes

Relation to Prior Work: I don't have a judgement because it’s out of my expertise domain.

Reproducibility: No

Additional Feedback:


Review 4

Summary and Contributions: The work filled an important gap for molecular design and optimization by considering the synthesis pathways of the molecules into the process, which is a prerequisite for experimental testing. Several contributions are made includes: 1. Representing synthesis pathway as DAG and serialzation of the it so that a RNN can be used. 2. A new message passing procedure for atoms in the molecules as well as nodes in the DAG, so that encoding from DAG to latent vector is possible. 3. Competitive results comparing to other baselines.

Strengths: 1. It filled an important application gap for molecular design and optimization. 2. Many technical contributions are made to use synthesis pathway. 3. I believe it will be an important work for molecule design and optimization community.

Weaknesses: Incorporating synthesis pathways into molecule design and optimization makes sense to me. But that comes with extra complexity. Apart from good empirical results the authors have shown, can you also comment on the complexity regrading to implementation, training time, inference time, especially considering JT-VAE is better on 3/5 metrics in Table 1.

Correctness: Yes.

Clarity: The paper is very well written.

Relation to Prior Work: Yes.

Reproducibility: Yes

Additional Feedback: There are many modelling choices and some ablation study or more detailed study on those components will shad more lights on method. For example: in the action embedding, GNN is preferred instead of fingerprint because other work shows GNN perform well. But in your setting, is it really true? If so, how much performance down-gradation will be observed? If the difference is limited, Fingerprint is so much simpler to use. ---After reading the author's feedback--- I am keeping the accept suggestion.

[Author Response · NeurIPS 2020]

We would like to thank the reviewers for their thoughtful comments. We are pleased that all four reviewers recommend acceptance, with R1 noting that they *"do not see any weaknesses with the current work"*. Furthermore, we are glad to hear that the reviewers are happy with the correctness of our work, the clarity (both R3 and R4 in particular said our paper was *"very well written"*), and the references to previous work (R2 stating that this was *"quite comprehensive"*). Moreover, we appreciate the reviewers highlighting the importance of our problem (R3: *"The paper aims to tackle a very interesting problem"*, R4: *"It filled an important application gap "*), the benefits of our approach (R2: *"proposed architecture to generate synthesis DAGs is very interesting"*, R4: *"many technical contributions"*), and the *"exciting potential extensions"* (R1). We respond to the individual specific questions from the reviewers below:

**Reviewer 1** *[.. caveat on imperfect MT and CASP oracles ..]* This is a good point; we are happy to add a statement to this effect in the paper to make it clearer to those less familiar with this area. An important point here is that our model is agnostic about which particular reaction predictor we use, so one could use a template based approach to achieve higher precision at the expense of lower recall. Being able to use any reaction prediction model also means that our method can take advantage of the future, independent development of these models to obtain improved performance.

*[..on formatting of references..]* Thank you for spotting this, this has now been fixed!

*[.. on reconstruction ..]* We are happy to bring this from the Appendix into the main text, thanks for the suggestion. As an alternative to greedy decoding we have also run an alternative where we sample 100 times from the model for each example and then resort on predicted probability. This obtains 66.2% accuracy.

**Reviewer 2** *[.. on quality of reaction predictor ..]* We shall add a comment on this issue as you suggest (see also our response to R1 above). Currently, if the reaction predictor suggests no products we select one of the input reactants to act as the product of the reaction – note however that if this happened frequently we would get very poor novelty results.

*[.. font size in figures..]* We shall increase the font size in the figures as per your suggestion – thanks for this.

*[.. dataset size .. ]* You bring up an interesting point. Our dataset size of 72000 does not appear to be a significant constraint so far in our experiments, as evidenced especially on the Valsartan optimization task: our approach was still able to find good molecules for this optimization task, even though the training set had no good molecules. However, we definitely agree it would be interesting to explore larger DAG datasets in future work! We believe these could either be extracted from large proprietary reaction datasets such as Reaxys or Pistachio or perhaps generated synthetically.

**Reviewer 3** *[3 (1)]* To deal with different possible orderings of the DAG serialization we randomly sample an ordering. As described in the caption to Figure 2, we also start at building blocks that are furthest from the final molecule node. This reduces the distance between introducing a building block and first using it to form a product molecule.

*[3 (2)]* Yes, the model could produce a DAG which is a sub-graph of another DAG by choosing to make a final product instead of an intermediate product for one of the product nodes. Although, perhaps pedantically, we would probably still class these as in-distribution as the model could have been trained on two such DAGs at training time.

**Reviewer 4** *[.. Training and inference times ..]* The training set DAGs have an average of 4.6 nodes, with the final molecules containing an average of 20.5 heavy atoms and 21.7 bonds (between heavy atoms). The average number of actions required to construct these DAGs is 11, as each reactant often contributes several atoms and bonds to the product. Using a NVIDIA K80 GPU it takes $\approx$ 7 mins to run a training epoch for DoG-AE ($\approx$ 0.4 secs per batch of 64). At inference time, where we do not initially have access to the complete sequence, we usually run larger batches, due to the fixed costs and latency of communicating with a Molecular Transformer server. It takes $\approx$ 29 secs to carry out *per batch of 200 DAGs* (and $\approx$ 12 secs per batch of 64 DAGs). Our approach could be sped up by using faster reaction predictors, although note that it still compares very favorably to full scale synthesis planning which takes on the order of 10 secs – 1 minute *per molecule*. Thank you for your comment, we shall add these details to the paper.

*[.. GNNs versus fingerprints..]* In this work we use GNNs due to their improved performance over fingerprints in previous work. GNNs are also popular in reaction prediction (e.g. [45,46]). It is worth pointing out that our GNN implementation is fairly lightweight: it only requires $\approx$ 73k parameters[1], the same weights are used in every message passing step, and the GNN/weights are shared between the encoder and decoder in the DoG-AE model. As we focused on the concept of DAG generation in this paper, and found performance to be sufficient, we did not perform extensive hyperparameter/architecture search. In production use, a fingerprint + MLP (as you reasonably suggest) or a different GNN/message-passing scheme, or perhaps even a concatenation of fingerprint and GNN features (shown to sometimes work well in [Yang et al., 2019, Fig. 16]) may be preferred, depending on performance/compute requirements.

Yang K, Swanson K, Jin W, et al. (2019) Analyzing Learned Molecular Representations for Property Prediction. Journal of chemical information and modeling 59(8): 3370–3388.

## Footnotes

[1]Note a learnt linear projection from 2048 dimensional molecular fingerprints to 50 dimensional embeddings would require $\approx$ 102k parameters in comparison.


[Meta-Review · NeurIPS 2020]

The reviewers find the presented method to be novel, important, and with compelling experimental evidence to support it's impact.